# Synthetic Routes to Coumarin(Benzopyrone)-Fused Five-Membered Aromatic Heterocycles Built on the α-Pyrone Moiety. Part 1: Five-Membered Aromatic Rings with One Heteroatom

**DOI:** 10.3390/molecules26020483

**Published:** 2021-01-18

**Authors:** Eslam Reda El-Sawy, Ahmed Bakr Abdelwahab, Gilbert Kirsch

**Affiliations:** 1National Research Centre, Chemistry of Natural Compounds Department, Dokki-Cairo 12622, Egypt; eslamelsawy@gmail.com; 2Plant Advanced Technologies (PAT), 54500 Vandoeuvre-les-Nancy, France; ahm@plantadvanced.com; 3Laboratoire Lorrain de Chimie Moléculaire (L.2.C.M.), Université de Lorraine, 57050 Metz, France

**Keywords:** coumarins, benzopyrones, five-membered aromatic heterocycles, furan, pyrrole, thiophene, selenophen

## Abstract

This review gives an up-to-date overview of the different ways (routes) to the synthesis of coumarin(benzopyrone)-fused, five-membered aromatic heterocycles with one heteroatom, built on the pyrone moiety. Covering 1966 to 2020.

## 1. Introduction

Coumarins are a family of benzopyrones (1,2-benzopyrones or 2*H*-[1]benzopyran-2-ones), which represent an important family of oxygen-containing heterocycles, widely distributed in nature [1,2,3,4]. Coumarins display a broad range of biological and pharmacological activities, [5,6] such as antiviral [7,8,9,10], anticancer [11,12,13], antimicrobial [14,15], and antioxidant [16,17,18] activities. On the other hand, coumarin represents an ingredient in perfumes [19], cosmetics [20], and as industrial additives [21,22]. Furthermore, coumarins play a pivotal role in science and technology as fluorescent sensors, mainly due to their interesting light-emissive characteristics, which are often responsive to the environment [23,24,25,26]. The coumarin (benzopyrane)-fused, membered aromatic heterocycles built on the α-pyrone moiety are an important scaffold. The only fused heterocycle with an α-pyrone moiety of coumarin that can be found in nature is the furan ring. One example is the naturally occurring furan 4*H*-furo[3,2-*c*]benzopyran-4-one, which provides the main core of many natural compounds of so-called coumestans. These comestans include coumestan, wedelolactone, and coumestrol. The coumestans are found in a variety of plant species that are commonly used in traditional medicine [27].



In order to enrich the limited versatility of the structures found in nature, synthesis of coumarin (benzopyrane)-fused, membered aromatic heterocycles has received considerable attention, including numerous reported routes.

This review gives an up-to-date overview of the different ways (routes) to the synthesis of benzopyrone-fused, five-membered aromatic heterocycles with one heteroatom, built on the pyrone moiety, from 1966 to 2020. Our main interest in this current work is to describe the components that have one heteroatom in an alicyclic-fused ring with the pyrone part of coumarin. The synthetic pathway of the investigated scaffold has provided systems containing oxygen, nitrogen, sulfur, and selenium in their core structure. The last heteroatom is less described in the output of the synthetic efforts. The fused heterocycles that contain more than one heteroatom will be detailed in the next part, which we intend to publish in the future.

Many strategies have been developed for the synthesis of the fused, five-membered aromatic heterocycle-benzopyran-4-ones. There are two main approaches to constructing these skeletons: five-membered, aromatic heterocycle construction, and pyrone-ring construction.

## 2. Synthesis of Benzopyrone-Fused, Five-Membered Aromatic Heterocycles

### 2.1. Five-Membered Aromatic Rings with One Heteroatom

#### 2.1.1. Furans

Furobenzopyrone (or furocoumarins) comprises an important class of coumarins found in a wide variety of plants, particularly in the carrot (*Apiaceae/Umbelliferae*), legume (*Fabaceae*), and citrus families (*Rutaceae*) [27]. The chemical structure of furobenzopyrone (furocoumarins) consists of a furan ring fused with coumarin. The fusion of the furan ring to the α-pyrone moiety of coumarin forms the core structure of the three most common isomers, viz. 4*H*-furo[2,3-*c*]chromen(benzopyran)-4-one, 4*H*-furo[3,4-*c*]chromen(benzopyran)-4-one, and 4*H*-furo[3,2-*c*]chromen (benzopyran)-4-one (Figure 1).

##### 4*H*-Furo[2,3-*c*]benzopyran-4-one


Furan Construction


The basic building block for the formation of 4*H*-furo[2,3-*c*]benzopyran-4-one is the 3-hydroxycoumarin (**1**) [28,29]. Pandya and coworkers [30] developed a method to synthesize some 4*H*-furo[2,3-*c*]benzopyran-4-ones starting with 3-hydroxycoumarin using the Nef reaction. Thus, the reaction of 3-hydroxycoumarin (**1**) with various 2-aryl-1-nitro ethenes **2a**,**b**, in the presence of piperidine and methanol as a solvent, followed the Nef reaction condition and afforded a series of 1-aryl-furo[2,3-*c*]benzopyran-4-ones **3a**,**b** and 1-phenyl-2-methyl-furo[2,3-*c*]benzopyran-4-one (**4**), respectively (Scheme 1). The formation of these products was explained by the reaction mechanism (Scheme 1).


Pyrone Construction


Dong et al., 2020 developed a novel and facile rhodium(III)-catalyzed process of sulfoxonium ylide (**5**) with hydroquinone (**6**). The carbonyl in the sulfoxonium ylide assisted the ortho-C–H functionalization of the sulfoxonium ylide, followed by intramolecular annulation with hydroquinone to afford 8-hydroxy-4*H*-furo[2,3-*c*]benzopyran-4-one (**7**) (Scheme 2) [31].

##### 4*H*-Furo[3,4-*c*]benzopyran-4-one


Furan and Pyrone Construction


In the literature, a large number of reports described the synthesis of 4*H*-furo[2,3-*c*] and 4*H*-furo[3,2-*c*]benzopyran-4-ones, while synthesis of the 4*H*-furo[3,4-*c*]benzopyran-4-one was reported by only one study, that of Brahmbhatt and his coworkers [32]. The first 4*H*-furo[3,4-*c*]benzopyran-4-ones (**10)** was synthesized by the demethylation–cyclization reaction of intermediates, 3-substituted-4-ethoxycarbonyl furans **9** (Scheme 3). For the demethylation and in situ lactonization steps, several reagents were tried, of which pyridine hydrochloride and HBr in acetic acid were found to be the most promising.

##### 4*H*-Furo[3,2-*c*]benzopyran-4-one


Furan Construction


A wide range of research has demonstrated that 4-hydroxycoumarin is the key compound for the synthesis of 4*H*-furo[3,2-*c*]benzopyran-4-ones, which can readily react with the C=C bond of the alkene, or the C≡C bond of the alkyne [33,34,35,36,37].

Reisch reported the condensation of 4-hydroxycoumarin (**11**) with 1-phenyl-2-propyn-1-ol (**12**) under acidic conditions (a mixture of glacial acetic and concentrated sulfuric acid) to deliver the corresponding 2-methyl-3-phenylfuro[3,2-*c*]benzopyran-4-one (**13**) (Scheme 4) [38].

A few studies employed the aliphatic aldehydes as building blocks with 4-hydroxycoumarin (**11**) to synthesize 4*H*-furo[3,2-*c*]benzopyran-4-ones [25,39]. This method was ineffective as it gave a poor yield as well as a mixture of 2,3-dihydrofuran, 4*H*-furo[3,2-*c*]benzopyran-4-ones, and 4*H*-furo[3,2-*c*]benzopyran-4-ones [39]. Conversely, in the case of using the aromatic aldehyde as a building block, the 4*H*-furo[3,2-*c*]benzopyran-4-one was obtained [40]. Kadam et al. developed atom-efficient multicomponent reactions (MCRs) and step-efficient, one-pot synthesis of 3-(4-bromophenyl)-2-(cyclohexylamino)-4*H*-furo[3,2-*c*]benzopyran-4-one (**16**) using 4-hydroxycoumarin (**11**) with 4-bromobenzaldehyde (**14**) and cyclohexyl isocyanide (**15**) as an alkylene source (Scheme 5) [40].

4-Hydroxycoumarin derivatives have received significant attention from researchers, as these derivatives possess 1,3-dicarbonyl systems. It allows for the easy generation of α,α′-dicarbonyl radicals, which can be readily added to the C=C bond of the alkene [41]. The first example of this reaction was described in 1998, by Lee and his coworkers. They reported an efficient way to prepare 4*H*-furo[3,2-*c*]benzopyran-4-ones **19** by Ag_2_CO_3_/celite (Fetizon’s reagent)-mediated oxidative cycloaddition of 4-hydroxycoumarin **17** to olefins, such as vinyl sulfide and phenyl propenyl sulfide. The resulting dihydrofuro[3,2-*c*]benzopyran-4-ones **18** was treated by sodium periodate in aqueous methanol to form the corresponding sulfoxides, which, upon refluxing with pyridine in carbon tetrachloride, directly delivered the 4*H*-furo[3,2-*c*]benzopyran-4-one **19** in good yields (Scheme 6) [41].

Recently, different catalytic methodologies have been developed for the synthesis of 2*H*-chromenes, and they are based on three main approaches: catalysis with (transition) metals, metal-free Brønsted catalysis, and Lewis acid/base catalysis, which includes examples of nonenantioselective organocatalysis and enantioselective organocatalysis [42,43,44]. Alkynes have been widely employed as building blocks for this reaction in most cases.

To date, different transition metal (Au, Pt, and Cu) catalyzed/mediated methodologies for benzopyrane synthesis have been reported [27,42,45,46]. Cheng and Hu described a one-pot cascade of an addition/cyclization/oxidation sequence using CuCl_2_ as the oxidant and CH_3_SO_3_H as the acid for regioselective synthesis of 2-substituted-4*H*-furo[3,2-*c*]benzopyran-4-ones **22** from the substituted 3-alkynyl-4*H*-benzopyran-4-one **20** (Scheme 7) [47]. This strategy included the CH_3_SO_3_H-acid-catalyzed construction of the furan ring, followed by oxidation of **21** with CuCl_2_ (Scheme 7) [47]. When the reaction was carried out in the presence of a catalytic amount of CuCl as a Lewis acid and atmospheric oxygen as an oxidative reagent, compound **22** was provided directly. On the other hand, the presence of 10% CuBr and an excess of CuCl_2_ as the oxidant afforded the corresponding 3-chloro-2-substituted- 4*H*-furo[3,2-*c*]benzopyran-4-ones **23** (Scheme 7) [48].

Brønsted-acid-catalyzed propargylations of several organic substrates, including 1,3-dicarbonyl compounds, with alkynols have been reported [49]. In most cases, the acid catalyst is required to promote the propargylation process efficiently. Zhou and coworkers developed a one-pot Yb(OTf)_3_ propargylation–cycloisomerization sequence of 4-hydroxycoumarin (**11**) with the propargylic alcohol (**24**) for the synthesis of a 2-benzyl-3- phenyl-4*H*-furo[3,2-*c*]chromen-4-one (**25**) skeleton using Yb(OTf)_3_ as a Lewis acid (Scheme 8) [50].

Similarly, 4*H*-furo[3,2-*c*] benzopyran-4-one formation reactions proceeded in higher yields and in a one-pot manner, employing a catalytic system composed of the 16-electron allyl–ruthenium(II) complex [Ru(η3-2-C_3_H_4_Me)(CO)(dppf)][SbF_6_] (dppf=1,1′-bis(diphenylphosphino)ferrocene) and trifluoroacetic acid (TFA) in the reaction of 4-hydroxycoumarin (**11**), with 1-(4-methoxyphenyl)-2-propyn-1-ol (**26**) as an example. The 4*H*-furo[3,2-*c*]benzopyran-4-one (**27**) was synthesized with a 72% yield (Scheme 9) [50,51,52].

Extensive work has been done to investigate the utility of an aryl alkynyl ether as a furan substrate, instead of arylalkynol, in the synthesis of 4*H*-furo[3,2-*c*]benzopyran-4-one [29,35]. The treatment of 3-iodo-4-methoxycoumarin (**28**) with phenylacetylene by means of sequential Sonogashira C–C coupling conditions resulted in a high-yield formation of the 4*H*-furo[3,2-*c*]benzopyran-4-one (**30**) (Scheme 10) [53]. In this reaction, the triethylamine was used as a base to induce the S_N_2-type demethylation of the Sonogashira coupling product, followed by an intramolecular attack of the enolate onto the cuprohalide π-complex of the triple bond (Scheme 10).

As a follow-up to this type of reaction, a novel and rapid assembly of an interesting class of 4*H*-furo[3,2-*c*]benzopyran-4-ones, **33**, was successfully achieved using a one-pot sequential coupling/cyclization strategy with 3-bromo-4-acetoxycoumarins **31** and dialkynlzincs **32** prepared in situ as reactive acetylides in transition-metal-catalyzed crosscoupling. The cascade transformation relies on palladium/copper-catalyzed alkynylation and intramolecular hydroalkoxylation (Scheme 11) [54].

A transition-metal-free approach was developed to achieve 4-*H*-furo[3,2-*c*]benzopyran-4-ones via an iodine-promoted one-pot cyclization between 4-hydroxycoumarins **34** and acetophenones **35**. The transformation spontaneously proceeded to produce (**36**) in the presence of NH_4_OAc. The possible reaction mechanism suggested for the iodine-promoted one-pot cyclization is depicted (Scheme 12) [55].

Additionally, Traven et al. [56] provided a new short way for the synthesis of 4*H*-furo- [3,2-*c*]benzopyran-4-one, employing the Fries rearrangement of 4-chloroacetoxycoumarin (**37**) to yield two products, namely 3-chloroacetyl4-hydroxycoumarin (**38**) and dihydrofuro[2,3-*c*]coumarin-3-one (**39**), in the ratio of 2:1. Compound (**38**), which underwent cyclization, led to the formation of (**39**). The latter, under reduction and dehydration conditions, afforded 4*H*-furo[3,2-*c*]chromen-4-one (**41**) (Scheme 13). A closely related reaction that allowed for the preparation of (**41**) was developed by Majumdar and Bhattacharyya [57], following a similar procedure but using chloroacetaldehyde instead of chloroactylchloride in the presence of aqueous potassium carbonate to give 3-hydroxy-2,3- dihydrofuro[3,2-*c*]benzopyran-4-one (**40**), which upon treatment with aqueous hydrochloric acid provided 4*H*-furo[3,2-*c*]benzopyran-4-one (**41**) with 72% yield (Scheme 13).


Pyrone Construction


Recently, much effort has been devoted to the development of oxidative intramolecular C–O bond-forming cyclization reactions for the synthesis of bioactive benzopyranones. These methods are limited to being used with arenes building blocks [58,59,60]. Fu et al. reported a ligand-enabled, site-selective carboxylation of 2-(furan-3-yl)phenols **42** under the atmospheric pressure of CO_2_. It was performed through an Rh(ii)-catalyzed C–H bond activation, assisted by the ligand chelation of the phenolic hydroxyl group to afford 4*H*-furo[3,2-*c*]benzopyran-4-ones **43** (Scheme 14) [61]. This reaction indicates the role of phosphine ligands in combination with Rh_2_(OAc)_4_ in promoting the reactivity and the selectivity during C–H carboxylation. The right choice of a suitable basic catalyst is an additional critical point.

#### 2.1.2. Pyrroles

Fusion of the pyrrole ring with the pyrone ring of coumarin (benzopyrane) leads to three structural isomers, viz. chromeno[3,4-*b*]pyrrol-4(*3H*)-one, chromeno[3,4-*c*]pyrrol-4(*2H*)-one, and chromeno[4,3-*b*]pyrrol-4(*1H*)-one (Figure 2).

##### 3*H*,4*H*[1]Benzopyrano[3,4-*b*]pyrrol-4-one


Pyrrole Construction


The 3-Aminocoumarin (**44**) is considered the starting compound for the preparation of fused *3H,4H*[1]benzopyrano[3,4-*b*]pyrrol-4-ones. The amino group represents the key moiety of this cyclization process in the reaction with different reagents [30,62,63,64]. Compound **45** was prepared by the reaction of 3-aminocoumarin (**44**) with different carbonyls via Fischer indole synthesis after being diazotized and reduced to coumarin-3-yl-hydrazine [62]. The compound 1-Aryloxy-4-chlorobut-2-ynes reacted with 3-aminocoumarin (**44**) to afford **47** through amino-Claisen rearrangement [64]. Condensation of (**44**) with α-halo ketones, followed by cyclization catalyzed by TFA, led to the formation of **49** [63], while **50** was prepared under Nef conditions using 2-aryl-1- nitro ethenes [30] (Scheme 15).

##### 3*H*,4*H*[1]Benzopyrano[3,4-*e*]pyrrol-4-one


Pyrrole Construction


A one-pot, three-component reaction of phenylsulphinyl-2*H*-benzopyran-2-one (**51**), phenylglycine (**52**), and benzaldehyde (**53**) led to the formation of 1,3-diphenyl[l]benzopyrano[3,4-*e*]pyrrol-4-one (**55**) (Scheme 16) [65].

Xue et al. developed an efficient and straightforward synthetic protocol for the preparation of [1]benzopyrano[3,4-*e*]pyrrol-4-ones **59** through FeCl_3_-promoted, three-component reactions between substituted 2-(2-nitrovinyl)phenols **58**, acetylene dicarboxylate (**57**), and amines **58** (Scheme 17) [66]. This reaction involved the sequential FeCl_3_-mediated nucleophilic addition of acetylenedicarboxylates, amines, and 2-(2-nitrovinyl)phenols, following intramolecular transesterifcation to form a coumarin core. This strategy offers a complementary approach to substituted pyrrolo[3,4-*c*]coumarin compounds, with advantages that include a variety of cheap and readily available reactants and a wide range of substrates with dense or flexible substitution patterns [66].

Alizadeh et al. reported a sequential three-component reaction of salicylaldehydes **60**, β-keto esters 61, and *p*-toluenesulfonylmethyl isocyanide (TosMIC) (**62**) via [1,3] acyl shift to give 2-acyl[1] benzopyrano[3,4-*e*]pyrrol-4-ones **63** (Scheme 18) [67]. A simple workup procedure, mild reaction conditions, lack of side products, and good yields of 62%–95% are the main aspects of this method.

Recently, Khavasi and his coworkers investigated the reactivity, chemo-, region-, and diastreo-selectivity of *p*-toluenesulfonylmethyl isocyanide (TosMIC) (**62**) in Van Leusen-type [3 + 2] cycloaddition reactions with the 3-acetylcoumarins **64** to give [1]benzopyrano[3,4-*e*]pyrrol-4-ones **65** (Scheme 19) [68]. This method offers several advantages, such as being inexpensive, providing good to excellent yields, producing short reaction times, high atom economy, and ease of product isolation under catalyst-free conditions without any activation at ambient temperature.

##### [1]Benzopyrano[4,3-*b*]pyrrol-4-one


Pyrrole Construction


Many synthetic protocols have been reported for the synthesis of [l]benzopyrano [4,3-*b*]pyrrole-4(1*H*)-ones, including the reaction of β-nitroalkenes **67** and 4-phenylamino coumarins **66** under solvent-free conditions to afford **68** (Scheme 20) [69]. Moreover, the reaction of the 4-aminocoumarin (**69**), amines **70**, and glyoxal monohydrates **71** in the presence of nanocrystalline CuFe_2_O_4_ [70], or KHSO_4_, led to the formation of [l]benzopyrano [4,3-*b*]pyrrole-4(1*H*)-ones **72** (Scheme 21) [71]. The synthesis of 71 using nanocrystalline CuFe_2_O_4_ discloses a rapid, high-yielding, green synthetic protocol for a variety of chromeno[4,3-*b*]pyrrol-4(1*H*)-one derivatives by assembling the basic building blocks in an aqueous medium using nano CuFe_2_O_4_ as the efficient, magnetically recoverable catalyst [70].

Many articles have found that the 4-chlorocoumarin is the key compound for the preparation of various [l]benzopyrano[4,3-*b*]pyrrol-4-ones by Knorr- or Fischer–Fink-type reactions [72,73]. Albrola et al. indicated the preparation of N(*α*)-(2-oxo-2*H*-l-benzopyran-4-yl)Weinreb-α-aminoamides **75** from 4-chlorocoumarin (**73**) and different α-aminoacids **74**. The reaction of **75** with various organometallic compounds, followed by cyclization, led to the formation of [l]benzopyrano[4,3-*b*]pyrrol-4-ones **76** (Scheme 22) [74,75].

On the other hand, the 4-amino-2*H*-benzopyran-2-one is employed as a key compound for the preparation of [l]benzopyrano[4,3-*b*]pyrrol-4-one [76,77]. Peng et al. synthesized a series of [l]benzopyrano[4,3-*b*]pyrrol-4-ones **79** via a palladium-catalyzed oxidative annulation reaction of 4-amino-2*H*-benzopyran-2-ones **77**, with electron-withdrawing or electron-donating groups with different alkynes **78** (Scheme 23) [76]. The method utilizes simple and readily available enamines and alkynes, and employs direct Pd(II)-catalyzed oxidative annulation to synthesize [l]benzopyrano[4,3-*b*]pyrrol-4-ones in high yields of 72%–99%.

Recently, Yang et al. reported a one-pot, two-step reaction of 4-amino-2*H*-benzopyran-2-one (**69**) with arylglyoxal monohydrates 80 and *p*-toluenesulfonates **81** to afford a series of 3-alkoxy-substituted [l]benzopyrano[4,3-*b*]pyrrol-4-ones **82** and **83**, respectively (Scheme 24) [78]. On the other hand, Yahyavi et al. described the Knoevenagle treatment of the arylglyoxals **84** with active methylene compounds and consequently an iodine-activated Michael-type reaction with 4-aminocoumarin (**69**) in a one-pot manner to afford disubstituted [l]benzopyrano[4,3-*b*]pyrrol-4-ones **85** (Scheme 24) [79].

#### 2.1.3. Thiophenes

Fusion of the thiophene ring with the pyrone ring of coumarin(benzopyrane) leads to three structural isomers, viz. *4H*-thieno[2,3-*c*]chromen(benzopyran)-4-one, *4H*-thieno[3,4-*c*]chromen(benzopyran)-4-one, and *4H*-thieno[3,2-*c*] chromen(benzopyran)-4-one (Figure 3)

##### 4*H*-Thieno[2,3-*c*]benzopyran-4-one


Thiophene Construction


The one-pot cascade addition/condensation/intramolecular cyclization sequence of chromone (**86**) with ethyl 2-mercaptoacetate (**87**) using a complexing ligand 1,8-diazabicyclo[5.4.0]undec-7-ene (DBU) in 1,4-dioxane led to the formation of 4*H*-thieno[2,3-*c*]benzopyran-4-one (**88**) (Scheme 25) [80].


Pyrone Construction


The Suzuki–Miyaura cross coupling of bromoarylcarboxylates and *o*-hydroxy(methoxy)arylboronic acids is one of the methods that plays an important role in the preparation of 4*H*-thieno[2,3-*c*] **91** and 4*H*-thieno[3,4-*c*]benzopyran-4-ones **92** (Scheme 26) [81,82,83].

##### 4*H*-Thieno[3,4-*c*]benzopyran-4-one


Thiophene Construction


4*H*-thieno[3,4-*c*]benzopyran-4-ones (**94**) were meanly prepared through the Gewald reaction (Scheme 27) [84,85,86,87,88,89,90]. Low yields of 37% and 48% were observed using these methods.

Yu et al. created a new technique for the preparation of 4*H*-thieno[3,4-*c*] [1]benzopyran-4(4*H*)-ones **97** by applying a chemoselective reaction of thioamides **95** with α-bromoacetophenones **96** (Scheme 28) [91].

A rhodium-catalyzed intramolecular transannulation reaction of alkynyl thiadiazoles **98** provided 4*H*-thieno[3,4-*c*][1]benzopyran-4(4*H*)-ones **99**. A plausible reaction mechanism proposed for the Rh-catalyzed intramolecular transannulation of alkynyl thiadiazoles is outlined in (Scheme 29) [92].

##### 4*H*-Thieno[3,2-*c*][1]benzopyran-4(4H)-one


Thiophene Construction


Numerous articles state the use of 4-chloro-2-*H*-benzopyran-3-carboxaldehydes **101** as a key compound in the preparation of a series of 4*H*-thieno[3,2-*c*][1]benzopyran-4(4*H*)-ones. This compound was prepared by a Vilsmeier–Haack reaction, and the cyclization process was performed through a rection with thioglycolate or dithiane to produce **102** and **103**, respectively (Scheme 30) [25,26,93,94].


Pyrone Construction


The palladium-catalyzed oxidative carbonylation of 2-(thiophen-3-yl)phenol (**104**) under acid-base-free and mild conditions yielded the corresponding 4*H*-thieno[3,2-*c*][1]benzopyran-4(4*H*)-one (**105**) (Scheme 31) [58].

#### 2.1.4. Selenophene

The fusion of the selenophene ring with the pyrone ring of coumarin leads to two structural isomers, viz. *4H*-selenophen[2,3-*c*]chromen(benzopyran)-4-one and *4H*-selenophen[3,2-*c*]chromen(benzopyran)-4-one (Figure 4).

##### 4*H*-Selenophen[2,3-*c*] and [3,2-*c*]benzopyran-4-ones

A few articles have discussed the possibility of synthesizing the fused selenophen-chromen-(benzopyran)-4-one moiety. A simple method for the synthesis of substituted 4*H*-selenopheno[2,3-*c*] benzopyran-4-ones **108** is by the treatment of 3-ethynylcoumarins **107** with selenium (IV) oxide and concentrated hydrobromic acid at room temperature [16,95]. Similarly, 4*H*-selenopheno-[3,2-*c*]benzopyran-4-ones **111** was prepared under the same conditions using 4-ethynylcoumarins **110**. The alkenyl derivatives were obtained from bromocoumarin (**106**) and 4-(trifluoromethane-sulfonyl)coumarin (**109**) by Sonogashira coupling. All the reaction steps were carried out in situ from the starting materials and until the end product (Scheme 32) [16,95].

Additionally, Kirsch and his coworkers reported the synthesis of selenopheno[2,3-*c*]benzopyran-4-ones (**115**) via a multistep reaction. This reaction cascade started by Vilsmeier formylation of 4-hydroxycoumarine (**111**). The formylated product reacted with hydroxyl amine to afford (**113**) according to the reaction conditions, which subsequently transformed into 3-cyano-4-coumarinselenol (**114**), by refluxing with selenium and sodiumborohydride in ethanol (Scheme 33). It was the precursor of selenopheno[2,3-*c*]benzopyran-4-ones (**115**) as it was reactive towards a series of haloacids, such as chloroacetonitrile, ethyl chloroacetoacetate, and chloroacetamide [96].

In conclusion, since coumarins have versatile applications, the synthesis of different structures of the coumarin-based scaffold was attempted. Among all the heterocycles built on the α-pyrone moiety of coumarin, the furan ring is the only available structure in nature. Thus, it has inspired a lot of researchers to replace the oxygen atom with other heteroatoms. Wide varieties of heterocycles were constructed by a synthetic pathway to introduce furans, pyrroles, thiophenes, and selenophenes as a fused ring that is characterized by a single heteroatom to the α-pyrone moiety of coumarin. The fused heterocycles that contain more than one heteroatom will be described in the next part, which we intend to publish in the future.

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
