# Peer review of "Synthetic Routes to Coumarin(Benzopyrone)-Fused Five-Membered Aromatic Heterocycles Built on the α-Pyrone Moiety. Part 1: Five-Membered Aromatic Rings with One Heteroatom"

_molecules, 2021, doi:10.3390/molecules26020483_

Round 1

Reviewer 1 Report

Manuscript ID: Molecules-1071408

Manuscript title: “Synthetic routes to coumarin(benzopyrone)-fused five-membered aromatic heterocycles built on the α-pyrone moiety. Part 1: Five-membered aromatic rings with one heteroatom

By Eslam Reda El-Sawy, Ahmed Bakr Abdelwahab and Gilbert Kirsch

Dear Editor,

The paper “Synthetic routes to coumarin(benzopyrone)-fused five-membered aromatic heterocycles built on the α-pyrone moiety. Part 1: Five-membered aromatic rings with one heteroatom” is a review article that involves the synthetic routes of coumarin(benzopyrone) fused to pyrroles, furans, thiophenes and selenophenes comprising the literature of range 1966 to 2020.

The paper is well written but, some question should be answered. In my opinion the paper, after this, should be accepted.

  1. The references must be indicated before the end point or the comma in all text.
  2. In the page 2, line 43 should put [16-18] instead [16][17,18].
  3. The authors should include the yields of the target compounds obtention for all Schemes. Only to the compound 27 and 30 showed the yields obtained.
  4. In the page 4, line 108, the compound 12 is with name incorrect according the Scheme 4, should be 1-phenyl-2-propyn-1-ol.
  5. In the page 7, Scheme 12, the mechanism contains an error because the authors putted acid medium (H+) and the presence of benzoic acid. The mechanism is not correct. Please remake.

Author Response

I suggest the following modifications before this paper is suitable for publication.

  • Given this is the first part of the review, a more comprehensive introduction should be provided. At least the outline of the entire review should be included.

Reply: It was done as required

  • A brief title should be given to each Scheme, in front of 'Reagents and conditions: ...'

Reply: thank you for your suggestion: they were added

  • There are too many simply mistakes throughout this paper, which should be avoided. Taking Page 8 as an example, '2.1.1.3.1' should be '2.1.1.3.2'; '2.1.1' should be '2.1.2'; 'f) COCH2Cl'?

Reply: Thank you for pointing our mistakes: they were checked and corrected. We tracked all numbering of all titles, subtitles, and formulas of reagents and reformed them in the right way.     

Reviewer 2 Report

I suggest the following modifications before this paper is suitable for publication.

(1) Given this is the first part of the review, a more comprehensive introduction should be provided. At least the outline of the entire review should be included.

(2) A brief title should be given to each Scheme, in front of 'Reagents and conditions: ...'.

(3) There are too many simply mistakes throughout this paper, which should be avoided. Taking Page 8 as an example, '2.1.1.3.1' should be '2.1.1.3.2'; '2.1.1' should be '2.1.2'; 'f) COCH2Cl'?

Author Response

This review is devoted to synthetical aspects of coumarin derivatives in al temporal range (more than seventy years).  Coumarins are effectively compounds of large interest and this would make the review appealing.

  • The authors effectively report the main contributions to this field. However, each method is just briefly discussed (or almost not discussed) and some important data about the synthetical methods reported are missed.  For instance, the yields of the reactions shown in the schemes are not reported and they should be added to the review. Overall, there is no attempt to compare the different methods used to obtain similar products where possible, which should be useful to readers.

Reply: We enriched the methods by the number of outputs from the reaction with their yields. 

  • I would like to suggest to the authors to add some critical comparison, where possible, between the methods used for the synthesis of the different compounds. This would increase the scientific soundness of this review, making it suitable for publication.

Reply: Thank you for your suggestion. We enriched the discussion and referred to the advantage of the mentioned methods     

  • Concerning the title, I note that this review seems to represent the first reports of a series of paper on the subject (‘part I’). Considering that the authors are providing a literature overview and not a research work, they should clearly describe their work plan in the introduction.

Reply: The next part was referred to in the introduction and the conclusion

  • Finally, numbers of compounds (in bold in the Schemes), should be also reported in bold in the text.

Reply: We followed the instructions of Molecules Journal. Where the numbers are written (not bold) inside the text, on converse inside the schemes

Reviewer 3 Report

This review is devoted to synthetical aspects of coumarin derivatives in al temporal range (more than seventy years).  Coumarins are effectively compounds of large interest and this would make the review appealing.

The authors effectively report the main contributions to this field. However, each method is just briefly discussed (or almost not discussed) and some important data about the synthetical methods reported are missed.  For instance, the yields of the reactions shown in the schemes are not reported and they should be added to the review. Overall, there is no attempt to compare the different methods used to obtain similar products where possible, which should be useful to readers.

Therefore, I would like to suggest to the authors to add some critical comparison, where possible, between the methods used for the synthesis of the different compounds. This would increase the scientific soundness of this review, making it suitable for publication.   

Concerning the title, I note that this review seems to represent the first reports of a series of paper on the subject (‘part I’). Considering that the authors are providing a literature overview and not a research work, they should clearly describe their work plan in the introduction.

Finally, numbers of compounds (in bold in the Schemes), should be also reported in bold in the text.

Author Response

Molecules Manuscript ID: Molecules-1071408 Manuscript title: “Synthetic routes to coumarin(benzopyrone)-fused five-membered aromatic heterocycles built on the α-pyrone moiety. Part 1: Five-membered aromatic rings with one heteroatom” By Eslam Reda El-Sawy, Ahmed Bakr Abdelwahab and Gilbert Kirsch Dear Editor, The paper “Synthetic routes to coumarin(benzopyrone)-fused five-membered aromatic heterocycles built on the α-pyrone moiety. Part 1: Five-membered aromatic rings with one heteroatom” is a review article that involves the synthetic routes of coumarin(benzopyrone) fused to pyrroles, furans, thiophenes and selenophenes comprising the literature of range 1966 to 2020. The paper is well written but, some question should be answered. In my opinion the paper, after this, should be accepted.

  1. The references must be indicated before the end point or the comma in all text.

Reply: checked and corrected

  1. In the page 2, line 43 should put [16-18] instead [16][17,18].

Reply: checked and corrected

  1. The authors should include the yields of the target compounds obtained for all Schemes. Only to the compound 27 and 30 showed the yields obtained.

Reply: Thanks for your suggestion. All yield and number of the outputs were added to schemes

  1. In the page 4, line 108, the compound 12 is with name incorrect according the Scheme 4, should be 1-phenyl-2-propyn-1-ol.

Reply: Thank you for pointing out our mistake. The name was corrected

  1. In the page 7, Scheme 12, the mechanism contains an error because the authors putted acid medium (H+ ) and the presence of benzoic acid. The mechanism is not correct. Please remake.

Reply: Thank you for pointing out our mistake. The reaction mechanism was corrected and redrew

Round 2

Reviewer 2 Report

I would recommend this manuscript for publication given that the authors have revised it according to reviewers' suggestions.

Reviewer 3 Report

The paper has appropriately been revised and can be puslished as such.